# Effect of Breeding Season on Haemosporidian Infections in Domestic Chickens

**DOI:** 10.3390/vetsci9120681

**Published:** 2022-12-07

**Authors:** Nuela Manka’a Che-Ajuyo, Xiaodong Rao, Boye Liu, Zhuqing Deng, Lu Dong, Wei Liang

**Affiliations:** 1Ministry of Education Key Laboratory for Biodiversity and Ecological Engineering, College of Life Sciences, Beijing Normal University, Beijing 100875, China; 2Ministry of Education Key Laboratory for Ecology of Tropical Islands, Key Laboratory of Tropical Animal and Plant Ecology of Hainan Province, College of Life Sciences, Hainan Normal University, Haikou 571158, China; 3College of Forestry, Wuzhishan National Long Term Forest Ecosystem Monitoring Research Station, Hainan University, Haikou 570228, China; 4Shaanxi Institute of Zoology, Xi’an 710032, China

**Keywords:** blood parasite, breeding season, chicken, *Gallus gallus domesticus*, *Leucocytozoon*, *Plasmodium*

## Abstract

**Simple Summary:**

A total of 122 chickens (66 chickens during the breeding season and 56 chickens during the non-breeding season) from tropical Hainan Island, China were sampled to test for haemosporidian infections. We showed that chickens have a higher incidence of haemosporidian infection and a greater diversity of haemosporidian parasite lineages during the breeding season relative to the non-breeding season.

**Abstract:**

Reproduction is believed to contribute to the frequently observed seasonal cycles in parasite loads in many organisms, as an investment in reproduction by the host could result in a higher susceptibility to parasites. In this study, we examined the impact of breeding season on haemosporidian infection in free-range chickens (*Gallus gallus domesticus*). We sampled a total of 122 chickens (66 chickens during the breeding season of April 2017 and 56 chickens during the non-breeding season of January 2017) to test for haemosporidian infections. The result showed that 56 out of 66 chickens examined during the breeding season tested positive for parasites (84.8% parasite prevalence), whereas 39 out of 56 chickens tested positive for parasites during the non-breeding season (69.6% parasite prevalence). Moreover, among the 11 *Leucocytozoon* lineages and 2 *Plasmodium* lineages identified, the parasite lineages that infected chickens during the breeding season were more diversified than those that affected chickens during the non-breeding season. This study indicated that chickens have a higher incidence of haemosporidian infection and a greater diversity of haemosporidian parasite lineages during the breeding season relative to the non-breeding season.

## 1. Introduction

Parasites can be a potent and selective force that strongly influences both the abundance and distribution of their host species [1]. There is a wide range of variability in the fitness consequences of parasitic infections [2]. The most relevant evidence for increased susceptibility to parasitic infections comes from studies conducted at breeding sites. As a result of the energy demands of reproduction in birds, less energy is devoted to the immune function during the breeding season [3,4]. In many hosts residing in temperate zones, avian haemosporidian infections peak in the spring and summer, when environmental conditions permit increased vector activity and hosts devote a great deal of energy to reproduction. The proposed reasons for the seasonal peak in infections include changes in the behavior and physiology of the parasite, vector, and host; however, it is difficult to regard these changes as separate processes, given that many of them occur simultaneously in the temperate zone.

Avian physiology changes dramatically during the spring breeding season [5,6]. Hormones regulate many of these processes, serving as broadscale messengers that deliver signals generated in the brain to target cells throughout the body. Several hormone systems could serve as parasite emergence cues. Hypothalamic-pituitary-gonadal-axis hormones (e.g., gonadotropins, androgens, and estrogens) mediate many of the cellular mechanisms involved in reproduction, including gonadal growth, reproductive behavior, and the allocation of resources to different processes [7]. As a result, these hormones may play a role in parasite emergence because reproduction in the majority of bird species temporally coincides with a high vector abundance in the spring. Testosterone has been shown to decrease immune function, thereby potentially enabling parasite re-emergence [8,9] or increasing infection severity [10]. In addition, glucocorticoids are frequently elevated during the early breeding season [11], which may indicate a high allostatic load or high energy turnover [12,13]. Energy investment in migration and reproduction in the spring may also result in trade-offs that compromise immune function, thereby allowing for the re-emergence of parasites [14,15,16].

Some host-specific factors that can affect parasite loads could be associated with the life history trade-offs that a host experiences owing to resource limitations, such as the energy available to an individual [17]. In particular, the trade-off between reproductive effort and parasite defense has been extensively studied. Various vertebrate species, such as female bats [18], squirrels [19,20], and deer mice [21], have shown increases in parasite loads during gestation, delays in the onset of breeding, as well as decreases in reproductive success as a result of parasite loads. Male sheep exhibit similar trade-offs between reproductive effort (measured by investment in sexually selected traits or parental effort) and parasite loads [22,23]. These studies indicate that investment in reproduction increases the susceptibility of the host to parasites.

In addition, it is believed that other factors based on the seasonality of reproduction in numerous organisms contribute to the commonly observed seasonal cycles in parasite loads [24]. These factors include the diversion of resources into reproduction and the selection of naive hosts; hence, they are thought to facilitate the successful propagation of parasites [25,26]. Some parasites appear to synchronize their reproduction cycles with those of their hosts; thus, peak loads are commonly observed during breeding seasons [27,28,29]. 

The fact that haemosporidian parasites require a vector to complete their life cycle increases the number of elements affecting their prevalence. Although some studies have demonstrated a limited or nonexistent association between climate and the prevalence of parasites [30], other studies have shown that climate change could lead to increases in vector populations; therefore, vector-borne infectious diseases may increase [31]. During warmer seasons, the probability of parasite infections of the host significantly increases, both because of vector availability and the secretion of sexual hormones that may alter the immune system and thereby facilitate the development of parasites within the vertebrate host [32]. 

In the present study, we aimed to determine the effects of the breeding season on haemosporidian infection in free-range chickens (*Gallus gallus domesticus*). We anticipated that parasite prevalence and the diversity of the lineages would be higher in chickens during the breeding season relative to comparable measurements in the non-breeding season.

## 2. Materials and Methods

A total of 122 free-range chickens (5 to 6 months old chickens) were sampled randomly (66 during the breeding season of April 2017 and 56 during the non-breeding season of January 2017) in Tunchang, tropical Hainan Island, southern China (19°30′38″ N, 110°10′29″ E).

Chickens were sampled based on their reproductive status (i.e., breeding or non-breeding) and were almost all from the same age group. In each group, there were both males and females (34 males from the breeding season and 32 females; and 28 males from the non-breeding season and 28 females). The following parameters were measured: body weight (BWt), body length (BLt), wing length (WL), tail length (TL), uropygial gland length (UGL), uropygial gland width (UGW), uropygial gland height (UGH), head length (HL), head breadth (HB), head height (HH), long spout (LSt), tarsal toe (TT), eye diameter (ED), eye short diameter (ESD), beak length (BL), beak width (BW), beak height (BH), wing span (WS), bite force 1 (BF1), bite force 2 (BF2), bite force 3 (BF3), average bite force (ABF), and maximum bite force (MBF). The BWt was measured to the nearest 0.01 g with a digital scale (KERN EMS 3000-2, DIYTrade International, Shenzhen, China). The HL, head width, and HH were measured to the nearest 0.01 mm using a digital caliper (Mitutoyo 500-196-30, Shenzhen Pantai Precision Instruments Co., Ltd. Guangdong, China). MBF was measured to the nearest 0.01 N with a micro-signal collector (NBIT-DUD-2404A, NBIT Inc., Nanjing, China) with a sensor probe (S2-200NHL-001, NBIT Inc., Nanjing, China). For each individual chicken, three values of BF were measured, with the highest value representing BF [33].

Blood samples of approximately 50–100 μL were collected from the brachial vein of chickens, stored in tubes with 95% ethanol, and transported to the Beijing Normal University for DNA extraction and subsequent laboratory processing. All ethanol was removed and DNA extracted from the blood samples using a TIANamp DNA kit (Tiangen, Beijing, China) following the protocols established by the manufacturer. Blood parasites were identified using a nested polymerase chain reaction (PCR) protocol based on the amplification of a 479-bp fragment of the mitochondrial cytochrome b (*cytb*) gene (derived from nested PCR) in accordance with the method described by [34]. Infections with *Plasmodium* and *Haemoproteus* were amplified with the outer primer pair HaemNFI/HaemNR3 (HaemNFI 5′-CATATATTAAGAGAAITATGGAG-3′ and HaemNR3 5′-ATAGAAAGATAAGAAATACCATTC-3′) and the nested pair HaemF/HaemR2 (HaemF 5′-ATGGTGCTTTCGATATATGCATG-3′ and HaemR2 5′-GCATTATCTGGATGTGATAATGGT-3′). *Leucocytozoon* infections were amplified with the outer primer pair HaemNFI/HaemNR3 and the nested primer pair HaemFL/HaemR2L (HaemFL 5′-ATGGTGTTTTAGATACTTACATT-3′ and HaemR2L 5′-CATTATCTGGATGAGATAATGGIGC-3′).

### Statistical Analysis

The data were statistically analyzed using the IBM statistics for Microsoft Windows, Statistical Package for Social Services (SPSS) version 25.0 (SPSS Inc., Chicago, IL, USA). Non-parametric tests were conducted for comparisons within the same group or between two independent groups using the one-sample test and the Mann–Whitney U test for independent sampling. Chi-square tests were conducted for grouping variables, and a one-way analysis of variance was used to separate the means of the measured body parameters.

The prevalence of haemosporidian parasites was estimated using a modified version of the formula used by Thrusfield [35]:

Percentage prevalence (%) = Frequency of a particular lineage in a population/Total number of frequencies of lineages observed in a population.

Sequences were aligned using Mega 7.0 [36] and blasted against the MalAvi database [37] for parasite identification. The haplotypes with a similarity to the MalAvi lineage that was greater than 98% were categorized as belonging to the same lineage. As evidenced by double peaks in chromatograms, mixed sequences were deemed co-infections [38]. For phylogenetic analysis, the resulting sequences were aligned using Mega 7.0 [36]. The evolutionary history was inferred by using the maximum likelihood estimation method based on the Tamura–Nei model [39]. The tree with the highest log likelihood (−22,114.74) is shown. Initial tree(s) for the heuristic search were automatically generated by applying Neighbor-Joining and Bio NJ algorithms to a matrix of pairwise distances estimated using the maximum composite likelihood approach with 1000 bootstraps and then selecting the topology with the highest log likelihood value. The codon positions that were included were 1st + 2nd + 3rd + Noncoding. All positions that contained incomplete and missing data were eliminated. The final dataset contained a total of 374 positions.

## 3. Results

Haemosporidian infection was more prevalent in chickens during the breeding season relative to the non-breeding season (Table 1). A total of 56 out of 66 chickens examined during the breeding season tested positive for the presence of parasites (84.8% parasite prevalence), whereas 39 out of 56 chickens tested positive during the non-breeding season (69.64% parasite prevalence). Moreover, additional parasite lineages infected chickens during the breeding season (Table 1 and Table 2).

The most prevalent lineages of haemosporidian parasites in chickens were the *Leucocytozoon* ZOBOR02 (30.36%) and Gallus08 (64.10%) during the non-breeding and breeding seasons, respectively. For the genus *Plasmodium*, there was a 1.76% prevalence of Gallus 01 in the chickens during the breeding season and a 5.13% prevalence of Gallus 04 in the chickens during the non-breeding season. There were five lineages of *Leucocytozoon* that were exclusively found in chickens during the breeding season, whereas two lineages of *Leucocytozoon* were solely discovered during the non-breeding season (Table 1). There were 17 and 16 cases of co-infections recorded in the chickens during the breeding and non-breeding seasons, respectively. Phylogenetic analysis revealed the diversity of the lineages (Figure 1).

In the comparison of the means for the measured body parameters of the chickens, significant differences were found in various distributions between the breeding season and non-breeding seasons (Table 3): BWt (df = 1, F= 27.018, *p* = 0.000); WL (df = 1, F = 13.822, *p* = 0.001); LSt (df = 1, F = 19.023, *p* = 0.001); TT (df = 1, F = 10.618, *p* = 0.001); HB (df = 1, F = 24.868, *p* = 0.000); HH (df = 1, F = 65.009, *p* = 0.000); ED (df = 1, F = 159.141, *p* = 0.000); ESD (df = 1, F = 92.838, *p* = 0.001); BL (df = 1, F = 8.132, *p* = 0.000); BW (df = 1, F = 12.007, *p* = 0.001); BH (df = 1, F = 20.644, *p* = 0.000); UGL (df = 1, F = 10.148, *p* = 0.002); WS (df = 1, F = 11.038, *p* = 0.001); UGH (df = 1, F = 10.901, *p* = 0.001); BF1 (df = 1, F = 48.936, *p* = 0.000); BF2 (df = 1, F = 86.609, *p* = 0.001); BF3 (df = 1, F = 87.893, *p* = 0.000); ABF (df = 1, F = 93.052, *p* = 0.000); and MBF (df = 1, F = 108.243, *p* = 0.001). Higher values of BWt, WL, eye size (i.e., ED and ESD), beak size (i.e., BL, BW, and BH), and uropygial gland size were observed in the chickens during the breeding season than those observed for chickens during the non-breeding season. However, higher values of ABF and MBF were observed in the chickens during the non-breeding season than those observed for chickens during the breeding season (Table 3).

## 4. Discussion

During the breeding season, most animals are susceptible to attacks by parasites, which may have deleterious effects [24]. Our results indicated that the prevalence of haemosporidians was generally greater during the breeding season than during the non-breeding season.

Avian reproduction is regulated by seasonal changes in corticosterone and sex hormones that are triggered by prolonged day lengths [40,41]. Thus, warmer seasons could increase the prevalence of parasites. Previous research has demonstrated that the prevalence of haemosporidian parasites can significantly increase during the breeding season owing to an abundance of vectors [42,43]. The prevalence of blood-borne parasites increases as the number of vectors in a community rises, leading to the transmission of blood-borne parasites among individuals [44]. Alternatively, the increased availability of vectors may result in a proliferation of blood parasites during the breeding season [45,46]. Furthermore, the chickens in the breeding season had longer WLs and BWs than those of the non-breeding season, which suggested that these birds were more susceptible to the physiological effects of parasitism during the breeding season [47]. In addition, the lineages of *Leucocytozoon* species that infected the chickens during the breeding season were more variable when compared to those that affected the chickens during the non-breeding season. It has been suggested that because of elevated stress levels, it is likely that various parasites may attack free-range chickens [48].

One limitation of our study was the absence of data on different flocks and years. Without this information, it was difficult to determine whether the absence of a particular haemosporidian genus in our sampled population was related to the sample size or lack of essential vectors. For example, we did not find any infections belonging to the *Haemoproteus* subgenus, despite it having been previously reported by Dunn et al. [49]. They conducted a 4-year study on yellowhammers (*Emberiza citronella*) and recorded cases of *Haemoproteus* infection, which is transmitted by louse flies (family Hippoboscidae) and primarily infects members of the family Columbiformes. Reinoso-Pérez et al. [50] also identified seasonal infections of *Haemoproteus* spp., *Leucocytozoon* spp., and *Plasmodium* spp. in various flocks of birds that were sampled in two different years. 

In conclusion, from the results of this study, we ascertained that haemosporidian parasites were highly prevalent in chickens during the breeding season. The prevalence rate of infections with this parasite and the diversity of its lineages were higher during the breeding season relative to the non-breeding season. We advocate that additional research be conducted to investigate the prevalence of haemosporidian infections across different geographical chicken populations. We further recommend that future studies link the physiological effects of haemosporidian infections in domestic chickens and wild birds.

## Figures and Tables

**Figure 1 vetsci-09-00681-f001:**
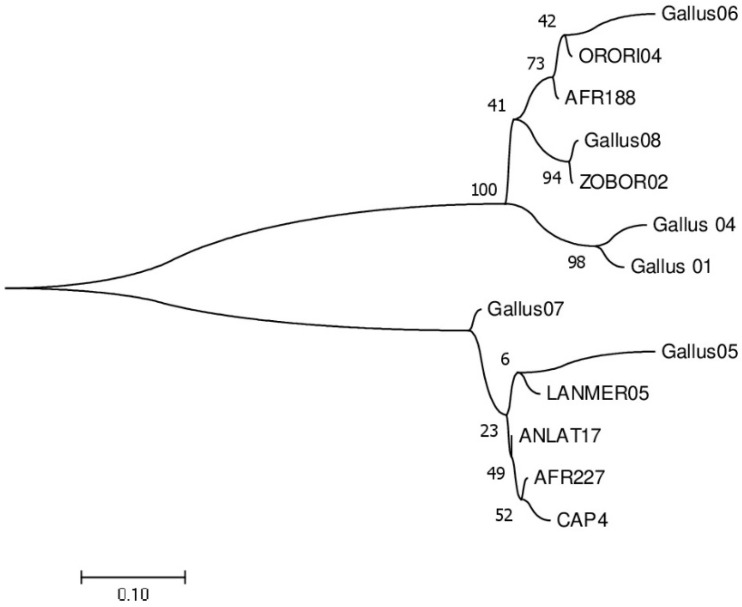
Phylogenetic relationships among the 13 haemosporidian lineages identified in this study. The number of branches represents bootstrap values derived from 1000 replicates.

**Table 1 vetsci-09-00681-t001:** Parasite lineage and prevalence in the breeding and non-breeding seasons.

Genus	Lineage	Breeding Season		Non-Breeding Season	
		Number of Chickens (n)	Prevalence (%)	Number of Chickens (n)	Prevalence (%)
Plasmodium	Gallus 01	1	1.76	0	0
Plasmodium	Gallus 04	0	0	2	5.13
Leucocytozoon	Gallus 05	7	12.50	2	5.13
Leucocytozoon	Gallus 06	6	10.71	1	0
Leucocytozoon	Gallus 07	5	8.93	0	0
Leucocytozoon	Gallus 08	9	16.07	25	64.10
Leucocytozoon	AFR188	4	7.14	0	0
Leucocytozoon	AFR227	2	3.57	0	0
Leucocytozoon	ANLAT17	2	3.57	0	0
Leucocytozoon	CAP4	3	5.36	0	0
Leucocytozoon	LANMER05	0	0	1	2.56
Leucocytozoon	ORORI04	0	0	2	5.13
Leucocytozoon	ZOBOR02	17	30.36	6	15.38
Total		56		39	

**Table 2 vetsci-09-00681-t002:** Number of male and female chickens sampled in the breeding and non-breeding seasons.

	Breeding Season		Non-Breeding Season	
	Number of Infected Chickens	Number of Uninfected Chickens	Number of Infected Chickens	Number of Uninfected Chickens
Male chickens	30	4	20	8
Female chickens	26	6	19	9
Total	56	10	39	17

**Table 3 vetsci-09-00681-t003:** Means (±SE) of body parameters of chickens in the breeding and non-breeding seasons.

Body Parameter	Breeding Season	Non-Breeding Season	Body Parameter	Breeding Season	Non-Breeding Season
BWt/g	960.23 ± 15.65 ^a^	834.43 ± 18.72 ^b^	ED/mm	12.52 ± 0.12 ^a^	10.53 ± 0.92 ^b^
BLt/mm	461.88 ± 8.44 ^a^	479.11 ± 6.93 ^a^	ESD/mm	10.02 ± 0.11 ^a^	8.59 ± 0.10 ^b^
WL/mm	215.48 ± 1.95 ^a^	206.58 ± 1.22 ^b^	BL/mm	18.38 ± 0.20 ^a^	17.61 ± 0.17 ^b^
TL/mm	195.97 ± 7.46 ^a^	200.63 ± 4.71 ^a^	BW/mm	19.22 ± 0.20 ^a^	18.34 ± 0.15 ^b^
UGL/mm	7.63 ± 0.16 ^a^	7.00 ± 0.10 ^b^	BH/mm	11.33 ± 0.16 ^a^	12.28 ± 0.12 ^b^
UGW/mm	11.36 ± 0.26 ^a^	10.52 ± 0.15 ^a^	WS/mm	7.73 ± 0.17 ^a^	6.96 ± 0.16 ^b^
UGH/mm	7.73 ± 0.17 ^a^	6.96 ± 0.16 ^b^	BF1/N	18.90 ± 0.76 ^a^	28.09 ± 1.11 ^b^
HL/mm	70.36 ± 0.68 ^a^	71.11 ± 0.68 ^a^	BF2/N	18.61 ± 0.76 ^a^	30.08 ± 1.00 ^b^
HB/mm	28.01 ± 0.22 ^a^	29.47 ± 0.19 ^b^	BF3/N	18.65 ± 0.70 ^a^	29.64 ± 0.98 ^b^
HH/mm	30.45 ± 0.27 ^a^	33.91 ± 0.34 ^b^	ABF/N	18.72 ± 0.66 ^a^	29.27 ± 0.88 ^b^
LSt/mm	31.46 ± 0.30 ^a^	29.62 ± 0.30 ^b^	MBF/N	21.14 ± 0.69 ^a^	33.31 ± 0.97 ^b^
TT/mm	69.66 ± 0.84 ^a^	73.88 ± 1.00 ^b^			

Note: The abbreviations in Table 3 refer to body weight (BWt), body length (BLt), wing length (WL), tail length (TL), uropygial gland length (UGL), uropygial gland width (UGW), uropygial gland height (UGH), head length (HL), head breadth (HB), head height (HH), long spout (LSt), tarsal toe (TT), eye diameter (ED), eye short diameter (ESD), beak length (BL), beak width (BW), beak height (BH), wing span (WS), bite force 1 (BF1), bite force 2 (BF2), bite force 3 (BF3), average bite force (ABF) and maximum bite force (MBF). Units of measurements: g for grams, mm for millimeters, and N for bite forces. Values in a row followed by the same superscripts are not significantly different (*p* ≥ 0.05).

## Data Availability

The datasets used in the present study are available from the corresponding author on reasonable request.

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
