# Peer review of "Effect of Breeding Season on Haemosporidian Infections in Domestic Chickens"

_vetsci, 2022, doi:10.3390/vetsci9120681_

Round 1

Reviewer 1 Report

Dear Authors, for comments and suggestions on paper, please refer to the attached file. Best regards.

Author Response

Thank you very much for your helpful comments. Now we have revised the MS following your and other three reviewers' suggestions. Please see attached "Response to comments" in PDF.

Reviewer 2 Report

The paper deals with a current and interesting topic. Nevertheless, the first fact that emerges from the work reading is that the variables involved in the different presence of haemosporidia in the two study groups are so numerous that it is very difficult to determine with certainty how these differences depend on the reproductive "status". In Loxia curvirostra, which reproduces both in winter and in summer, the hormonal status in the reproductive phase is the same, in the two seasons, but not the number of hemosporidial infections, meaning that the reproductive "status" is not the only predisposing factor for development of hemosporidia (Cornelius et al., 2014).

On lines 102 and 103 you state that the purpose of your work is to determine the effects of breeding "status" on haemosporidian infection in free-range chickens. in discussions you talk about breeding "season" (line 220-222-228-232-234-236-238-239-254-256) and don't talk about breeding "status". Moreover, you have said little about the different management of the two study groups: did they follow the same diet? Are there any additions during the reproductive phase? Is the sleep-wake period the same? What is the age of the chickens used? What weight does the relapse have on the results of the breeding period?

In conclusion, it is evident that in the reproductive "season" the cases of hemosporidial infection increase, but how much does the reproductive "status" influence the results obtained? Were you able to determine the effects of breeding "status" on haemosporidian infection in free-range chickens?

suggestion: line 2, 20 and 102 replace "status" with "season"

Author Response

Thank you very much for your kind comments. Now we have revised following your suggestions. Please find attached PDF.

Reviewer 3 Report

The results were few but interesting. I liked the introduction, which gives quite a broad overview of the host's breeding status on parasite development. 

In spite of mentioned, I found some major indistinctness in this manuscript.

1. What is the relationship between the prevalence of haemosporidian parasites and measurements of chickens? Why were these measurements taken and used in this research?

2. There was no data about vector abundance provided, so we can not make conclusions about them.

3. The obtained results did not convince that "Our findings indicate a clear correlation between breeding season, vector availability, and the prevalence of blood parasites..." (lines 233-234) It is not a correlation, because You have only 2 data (January and April of the same year) and one of them is greater compared with another.

Minor remarks.

What was the age of chickens?

Did You investigate the same chickens (in January and in April) of different?

I would suggest not using Fig. 1. There is no Genus and outgroup in this tree and the tree is not informative. It is also a repetition of Fig. 2.  Or You should explain the difference (about information provided) between Fig.1 and Fig. 2. 

Table 2 - it should be marked which measurements are statistically different and which are not. 

Line 154. "The haplotypes with greater than 98% similarity to the MalAvi lineage were categorized as belonging to the same lineage". To my knowledge, each cytb haplotype is a separate genetic lineage, o 98% is a too big difference.

Lines 236-237. "...chickens in the breeding season had longer WLs and BWs than those of the non-breeding season, which suggested that these birds were more susceptible to the physiological effects of parasitism during the breeding season..." Why these parameters are related?

This manuscript can be prepared as a short communication because in the common results about parasite prevalence are interesting. 

Author Response

Thank you very much for your helpful comments.

Now we have revised and please find attached PDF.

Reviewer 4 Report

Please see comments in the attached file.

Author Response

Thank you very much for your detailed suggestions. Now we have revised. Please find attached PDF.

Round 2

Reviewer 1 Report

The paper has been sufficiently improved and may be accepted without any further changes.